# Evaluation of the Inhibitory Potential of Casuarictin, an Ellagitannin Isolated from White Mangrove (*Laguncularia racemosa*) Leaves, on Snake Venom Secretory Phospholipase A2

**DOI:** 10.3390/md17070403

**Published:** 2019-07-08

**Authors:** Caroline Fabri Bittencourt Rodrigues, Marcelo José Pena Ferreira, Mariana Novo Belchor, Caroline R. C. Costa, Danielle P. Novaes, Adeilso Bispo dos Santos Junior, Cinthia I. Tamayose, Marcus Vinícius Terashima Pinho, Marcos Antonio de Oliveira, Marcos Hikari Toyama

**Affiliations:** 1Laboratório de Bioquímica e Biologia Molecular de Peptídeos (BIOMOLPEP), Instituto de Biociências, UNESP, Campus do Litoral Paulista, São Vicente 11330-900, São Paulo, Brazil; 2Laboratório de Herpetologia, Instituto Butantan, São Paulo 05503-900, São Paulo, Brazil; 3Departamento de Botânica, Instituto de Biociências, Universidade de São Paulo, São Paulo 05508-090, Brazil; 4Laboratório de Biologia Molecular Estrutural (LABIMES), Instituto de Biociências, UNESP, Campus do Litoral Paulista, São Vicente 11330-900, São Paulo, Brazil

**Keywords:** casuarictin, *Laguncularia racemosa*, secretory phospholipase A2, *Crotalus durissus terrificus*, enzymatic inhibition, anti-inflammatory, edema and myonecrosis

## Abstract

Ellagitannins constitute the largest group of hydrolyzable tannins of plants, and, from this group, casuarictin (Casu) was identified in some plant species. However, to our knowledge, no investigation of secretory phospholipase A2 (sPLA2) inhibition by Casu has been performed yet. Casuarictin was isolated by chromatography n-butanol (n-BuOH) partition of *Laguncularia racemosa* leaves. The pharmacological and biological effects of Casu were evaluated on isolated sPLA2 from the rattlesnake (*Crotalus durissus terrificus*) and using a plant bacterial strain. The compound was able to form a protein complex consisting of a stable sPLA2 + Casu complex. Analyses carried out with matrix-assisted laser desorption ionization-time-of-flight mass spectrometry (MALDI-TOF) revealed that the molecular mass of sPLA2 increased from 14,425.62 to 15,362.74 Da. The enzymatic activity of the sPLA2 + Casu complex was significantly lower than that of native sPLA2. Besides, molecular interactions of Casu with sPLA2 were able to virtually abolish the native edematogenic effect as well as myonecrosis induced by the protein when injected 10 min after sPLA2. Therefore, Casu may be considered a potential anti-inflammatory that can be used to treat edema and myonecrosis induced by serine-secreting phospholipase A2. In addition, the compound also showed great antimicrobial potential.

## 1. Introduction

Phospholipase A2 from snake venom, such as that isolated from *Crotalus durissus terrificus* (Cdt), shares structural similarity with mammalian secretory phospholipase A2 (sPLA2) and uses the same enzymatic and non-enzymatic mechanisms for its pharmacological actions [1,2]. Thus, the pharmacological activity of snake venom which produces phospholipase A2 can be driven by the interaction of the "pharmacological site" of this enzyme on the target cell and by the ability to generate arachidonic acid (AA) through its enzymatic activity [1,3]. The interaction of sPLA2 from venom with enzyme receptors, or the enzymatic capacity, involves complementarity in terms of charge, hydrophobicity, and other amino acid residues with the pharmacological or enzymatic site of sPLA2. Therefore, compounds capable of inhibiting the enzymatic activity of sPLA2 or interacting with the pharmacological site of this enzyme can significantly decrease its toxicity [4,5]. Mangroves represent a complex coastal ecosystem placed between marine and terrestrial ecosystems that is subjected to a tidal regime. In Brazil, mangroves occur on almost the whole coast and cover approximately 13,762 km^2^. Only four plant species are available in Brazilian mangroves: *Avicennia schaueriana*, *Avicennia germinans*, *Laguncularia racemosa,* and *Rhizophora mangle* [6]. 

In several cultures, mangrove plants have been used in folk medicine, and their extracts exhibit inhibitory activity against human, animal, and plant pathogens. From this ecosystem, secondary metabolites, such as alkaloids, polyphenols, steroids, and terpenoids, have been characterized, revealing toxicological, pharmacological, and ecological significance [7,8]. Polyphenolic compounds, such as glycosylated flavonoids, inhibit the activity of certain essential pharmacological enzymes such as thrombin and sPLA2 [8]. Other studies showed that mangrove plants and their parts, including roots, barks, and leaves, produce several chemical compounds, which are used in folk medicine with diverse applications such as antiparasitic and antimicrobial treatments, treatment of skin disease, pains, diarrhea, and inflammation [8,9,10,11,12]. Among these compounds, tannins are a large, heterogeneous, and diverse group of polyphenolic compounds found in various plant species, which provide protection against biotic and abiotic stressors. Depending on their chemical composition, tannins can be classified into two main groups: Condensed tannins and hydrolyzable tannins [13]. Mangrove plants exhibit both groups of tannins, which are found in all parts of the plant. For example, the presence of tannins in *L. racemosa* is found mainly in the bark; however, this group of compounds can also be found in leaves in a lower concentration [7,14,15]. Casuarictin (Casu) is a hydrolyzable ellagitannin with antioxidant and antiviral activities, and despite this compound constituting a major group in some plants, there are no descriptions of its presence in *L. racemosa*.

## 2. Results

### 2.1. Isolation, Structural Characterization, and Biological and Pharmacological Trials of Casuarictin, a Tannin from Laguncularia racemosa Leaves

Polar phases (ethyl acetate (EtOAc) and n-butanol (n-BuOH)) obtained from *L. racemosa* leaves were evaluated in the enzymatic activity of sPLA2 from Cdt, and the results are shown in Figure 1. It was observed that both phases significantly decreased the enzymatic activity of the enzyme, with the EtOAc phase (sPLA2 + EtOAc) showing greater inhibition capacity than the n-BuOH phase. Both fractions were inhibited in the same manner during the first 40 min of the enzymatic assay time course; however, the EtOAc phase clearly showed an irreversible enzymatic diminishing effect (Figure 1A). Figure 1B shows that sPLA2-treated phytobacteria previously incubated with the EtOAc phase reversed the native sPLA2 antibacterial activity, indirectly confirming the data of Figure 1A. Figure 1C,D shows that the EtOAc phase was able to significantly decrease the pharmacological activity of sPLA2 by decreasing edema and the myotoxic activity, i.e., the in vivo activities. Figure 1E,F shows the effects of administration of the BuOH partition phase 10 min after the injection of sPLA2, and this phase inhibited edema but was not able to inhibit myotoxic activity. These results show that the EtOAc phase revealed a better inhibition effect on enzymatic activity and a better pharmacological inhibition capacity than BuOH. EtOAc from a methanolic extract of *L. racemosa* was subjected to the Sephadex LH-20 column to separate components, yielding four groups. Group L1 revealed a major component with a retention time of 3.7 min. The ^1^H NMR spectra showed a singlet at δ 7.18 (^2^H), characteristic of a galloyl group, and four other singlets at δ 6.69, δ 6.57, δ 6.45, and δ 6.38, which were assignable to two hexahydroxydiphenoyl (HHDP) groups. Additionally, the sugar moiety was assigned by the four doublets at δ 3.22 (*J* = 3.8 Hz), δ 3.62 (*J* = 3.8 Hz), δ 3.52 (*J* = 2.9 Hz), and δ 3.26 (*J* = 2.9 Hz), a multiplet at δ 3.80 and a double–doublet at δ 4.01 (*J* = 9.0, 2.0 Hz), indicating the presence of glucopyranose. The ^13^C NMR spectra confirmed the presence of a galloyl and two HHDP groups esterifying the glucose moiety. HMBC correlations confirmed the galloyl unit at C-1 and two HHDP groups at C-2,C-3 and C-4,C-6. Therefore, the compound was identified as casuarictin 1-*O*-galloyl-2,3:4,6-di-*O*-(4,4′,5,5′,6,6′-hexahydroxydiphenoyl)-glucopyranose by comparison with the literature data [16]. Figure 2 shows the structure of casuarictin, the tannin isolated from *L. racemosa* (Figure 2).

### 2.2. Biochemical Characterization of Tannin and sPLA2 Interaction

HPLC SEC analysis (size exclusion chromatography) of sPLA2 and Casu showed that the tannin is able to interact with sPLA2 and form a molecular complex, which appears to be stable (Figure 3A). The complex remained stable after purification of the molecular complex (sPLA2:Casu) through fractionation with reverse phase chromatography (Figure 3B). Figure 3A shows that sPLA2 isolated from the Cdt snake venom exhibits a single peak eluted at 10.5 min, with an estimated molecular weight of 30 kDa. After sPLA2’s incubation with Casu, we observed a single peak with an estimated molecular mass of 13 kDa. PAGE-SDS analysis was used to calculate the molecular mass using a 12% resolution gel in acrylamide. In addition, reverse phase HPLC analysis of sPLA2 isolated from Cdt snake venom in native form or after treatment with Casu tannin (sPLA2:Casu) showed that native sPLA2 was eluted at 3 min into the chromatogram, whereas the sPLA2:Casu peak was eluted as a single peak at 42 min. The molecular analysis of sPLA2 and sPLA2 + Casu performed in matrix-assisted laser desorption ionization-time-of-flight mass spectrometry (MALDI-TOF) showed that the enzyme exhibits a molecular mass of 14,425.62 Da, whereas the sPLA2:Casu complex showed 15,362.74 Da. The results of analyses between sPLA2 and sPLA2:Casu showed a difference of 937 Da, which is the molecular mass of the tannin. These results show that the Casu tannin was able to form a stable complex with sPLA2, which probably involves strong non-covalent interactions such as hydrogen bonds and hydrophobic interactions. Figure 3D reveals that the chemical interaction of sPLA2 with Casu significantly decreased sPLA2 enzymatic activity. 

### 2.3. Effects of Casu on the Enzymatic and Pharmacological Activities of sPLA2

Figure 4 shows the effects of the treatment of sPLA2 with the tannin isolated from the aerial parts of *L. racemosa*. Native sPLA2 induced acute edema with a maximal edema peak of 108 ± 5 μL (*n* = 5, * *p* ≤ 0.05) at 30 min, subtracting the saline value (*n* = 5, * *p* ≤ 0.05). In animals treated with the tannin (200 mg/kg), the compound was injected intraperitoneally (100 μL/animal) after 10 min of sPLA2 injection, showing an edema peak of 64 ± 7 μL (*n* = 5, * *p* ≤ 0.05) at 30 min, subtracting the saline value (*n* = 5, * *p* ≤ 0.05). In addition, we used the flavonoid naringenin (Nar) as an anti-inflammatory control, which was administered under the same experimental conditions used for Casu (Figure 4A). The results presented in Figure 4A show that Casu significantly diminished the edema induced by sPLA2. Moreover, previous incubation of sPLA2 with *L. racemosa* tannin (sPLA2:Casu) showed an edema value of 36 ± 11 μL (*n* = 5, * *p* ≤ 0.05) at 30 min, subtracting the saline value (*n* = 5, * *p* ≤ 0.05), and the other edema values were close to the saline value. Myotoxicity was evaluated by the creatine kinase (CK) level and sPLA2 injection into muscle induced a significant increase in plasmatic CK levels (*n* = 5, * *p* ≤ 0.05). Casu showed a significant inhibitory or protective effect against myonecrosis induced by the enzyme, revealing a CK level close to that of saline (negative control) when injected for 10 min intraperitoneally (200 mg/kg; 100 μL/animal). This anti-myonecrotic effect of Casu tannin was supported by microscopy investigation. For animals that received sPLA2, consistent muscle disorganization was observed with swelling and diffuse muscle fibers. However, animals treated with Casu exhibited muscle fibers with great similarities to animals that received saline.

Otherwise, sPLA2 previously treated with *L. racemosa* tannin (sPLA2:Casu) induced an increase in CK plasmatic levels, similar to sPLA2. In Figure 4C, the effects of different substrate concentrations on the enzymatic activity of sPLA2 purified from Cdt venom can be observed. When Casu tannin was added to the enzymatic solution 10 min after the addition of sPLA2, the enzymatic activity was irreversibly inhibited, as shown by the enzymatic activity.

### 2.4. Antimicrobial Effects of Casu Tannin 

In addition, Casu decreased the bacterial growth rate expressed as colony formation units (CFUs in percent) in a dose-dependent manner. Figure 5 exhibits the antimicrobial effect of the tannin against two different phytobacteria, revealing that the growth rate of gram-positive bacteria (*Clavibacter michiganensis* pv. *michiganensis*) was strongly inhibited by this compound, whereas gram-negative bacteria (*Xanthomonas axonopodis* pv. *passiflorae*) were more resistant. Electronic transmission microscopy revealed that Casu was able to induce several bacterial wall destabilizations characterized by rupture and extension.

## 3. Discussion

Tannins are polyphenolic compounds found throughout the plant kingdom, which can be divided in two groups: Condensed tannins and hydrolyzable tannins. In mangrove plants, these compounds play a key role in defense against attacks by microorganisms and other predators [17,18]. Besides their defensive function, tannins from mangrove plants are essential for the maintenance and physiological sustentation of these plants, as mangroves are an ecosystem with an extreme environment due to the presence of high concentrations of salts [19]. These polyphenolic compounds have been employed for millennia in cosmetics, folk medicine, and food industry uses. Tannins have great potential for the protection of plants against insects, diseases, and herbivores. Additionally, several studies have revealed the potential of these compounds in the development of pharmaceuticals and the conservation of wood [20,21]. Likewise, these compounds are able to induce protein, carbohydrate, and metal ion precipitation, revealing their use as tools for the precipitation of proteins and other organic molecules [22]. Furthermore, tannins in general display several pharmacological actions including anti-inflammatory, antioxidant, blood clotting, antidiarrheal, and anti-adherent effects [17,23].

Casu isolated from different plant species such as *Carpinus tschonoskii* and strawberry is able to inhibit allergic reactions induced by BSA (bovine serum albumin) and leads to NF-kβ inhibition, exerting a gastric anti-inflammatory function [24,25]. However, the evaluation of Casu anti-inflammatory activity in the presence of sPLA2 as a pro-inflammatory agent is essential considering the fundamental role of this enzyme in the inflammatory process [26]. This study showed that Casu can inhibit the pro-inflammatory effect induced by sPLA2, strongly decreasing the enzymatic activity of the enzyme isolated from the Cdt total venom, both in vitro and in vivo.

In vitro, tannin was able to form a stable molecular complex, and SEC analyses showed that Casu was able to abolish sPLA2 homodimeric complex formation from approximately 30 to 13 kDa. Some electrophoresis and spectroscopic studies have suggested that the dimer-like structure in solution is similar to sPLA2s from other snake venoms. The presence of the dimeric form is an essential feature for the pharmacological and physiological function of sPLA2, but self-oligomerization of the enzyme seems to be essential for catalysis [27,28,29]. SEC and reverse phase HPLC showed that Casu chemically reacts with sPLA2 to inhibit the development of the dimeric form of this enzyme. Otherwise, this compound was able to diminish access of the substrate to the catalytic site, as shown in Figure 4C, which shows that the “switch-like” transition found for native sPLA2 disappeared after chemical treatment of sPLA2 with Casu in the enzymatic assay.

Considering these results, it is possible that the tannin decreases the enzymatic activity of sPLA2 from *C. durissus terrificus* through the inhibition of sPLA2 dimerization. sPLA2 hydrolyzes a wide range of phospholipid substrates depending on the isoform, and several studies have shown that catalytic activity plays a crucial role in triggering pharmacological activity [30,31]. Hence, drugs that inhibit the catalytic activity of sPLA2 may lead to a decrease in its pharmacological and biological activity [32,33]. Figure 2 shows Casu as a compound abundantly distributed in the BuOH and EtOAc partitions. This compound would be the agent responsible for strongly decreasing the enzymatic and pharmacological activity of sPLA2. In the case of the anti-inflammatory activity induced by the partitions, studies show that extracts of plants rich in tannins are able to inhibit the inflammatory process [27]. Therefore, the results observed for the tannin-rich partitions are compatible with data in the literature.

Figure 4 reveals that the tannin applied 10 min after the sPLA2 injection was able to significantly decrease the edema as well as the myotoxic activity induced by sPLA2. On the other hand, protection against edema and/or myotoxicity may also result from the antioxidant capacity of the compound [34,35]. Some studies have shown that small increases in AA can induce a cascade of reactions that culminates with the increase of reactive oxygen species. In addition, AA metabolism itself can also generate other reactive oxygen species [27]. Thus, the protective activity of Casu tannin on the course of pharmacological activity induced by sPLA2 could involve three mechanisms of action: Inhibition of the enzymatic activity of sPLA2, the antioxidant capacity of the compound, and the ability to form aggregates with proteins. In Figure 4A, we also show the effect of naringenin, which was used as the reference compound for anti-inflammatory activity. In addition, naringenin (Nar) also neutralizes the enzymatic and pharmacological effects induced by sPLA2 [36], and its anti-inflammatory activity also involves intracellular and extracellular mechanisms [37,38]. The Casu compound generally showed efficient and comparable anti-inflammatory activity to the control of naringenin, mainly after 60 min of the edema experiment. Additionally, regarding the anti-inflammatory effect of Casu purified from *L. racemosa*, this tannin exhibited significant antimicrobial activity against gram-positive and gram-negative bacteria, which suggests that this compound has potential therapeutic application.

## 4. Materials and Methods 

### 4.1. Materials and Animals

The crotoxin and sPLA2 used in this work were commercially purchased from Sigma Aldrich, St. Louis, MO 63103, USA (product numbers C3737 and P5910, respectively) and subjected to a fractionation process for final purification. Reagents for the experimental trials were purchased from Promega, Sigma-Aldrich, Fisher, Across, BioRad, and GE Healthcare Life Sciences. Chromatographic columns were purchased from Phenomenex (Torrance, CA, USA) or Sigma-Aldrich (Supelco, Darmstadt, Germany). Sephadex LH-20 (GE Healthcare) was used for column chromatographic separation, while silica gel 60 PF254 (Merck, Darmstadt, Germany) was used for analytical thin layer chromatography (TLC) (0.25 mm). ^1^H NMR and ^13^C NMR spectra were recorded at 300 and 75 MHz, respectively, in a Bruker DPX-300 spectrometer (Bruker, Billerica, MA, USA). CD3OD (Aldrich) was used as solvent and Tetramethyldisilazine (TMS) (Aldrich) as an internal standard. 

Chemical shifts are reported in δ units (ppm) and coupling constants (J) are reported in Hz. Analytical HPLC analysis was performed using a Dionex chromatograph model P680 (Thermo, Waltham, MA, USA) with a UV–Vis detector (model UVD 340U) (Thermo, Waltham, MA, USA) and a C-18 column (3.5 μm, 150 × 5 mm) with a flow rate of 0.6 mL.min^−1^ using H_2_O:HOAc 2% and MeOH as the eluent. Mass spectra were obtained on a Bruker Daltonics MicrOTOF mass (Bruker, Billerica, MA, USA), Spectrometer (Electrospray ionization (ESI-MS)) and Bruker Daltonics UltrOTOF mass spectrometer (ESI-MS/MS) (Bruker, Billerica, MA, USA). The animals used in the in vivo tests were Swiss female mice (20–30 g), from the Central Animal House of the State University of Campinas, with appropriate certification from the Animal Ethics Committee of Universidade de Campinas (UNICAMP) (413141), as well as the ethics committee of University São Paulo State, Biosciences Institute of the São Paulo Coast (UNESP IB-CLP), CEUA 002/2014, where the tests were carried out. The animals were kept in polypropylene cages and placed in a Controlled Animal Housing System, with photoperiod control, temperature controlled between 22 and 24 °C, and a ventilation system equipped with HEPA filters. Food and water were provided ad libitum.

### 4.2. Plant Material, Extraction, Isolation, and Identification of Compounds

Leaves from *L. racemosa* were collected in a mangrove in Praia Grande city, São Paulo, Brazil (23°59′15″ S, 46°24′18″ W). The voucher specimen was deposited at the Herbarium of the University of Santa Cecília under the number HUSC8256. Leaves were washed in distilled water, dried, and lyophilized. Powdered leaves of *L. racemosa* (39.4 g) were completely degreased with hexane (at room temperature). Sequentially, the plant material was exhaustively extracted with MeOH, yielding 29.4 g of a green extract after removal of the solvent under reduced pressure. The MeOH extract was resuspended in MeOH:H_2_O (1:2, v/v) and then partitioned successively with hexanes, dichloromethane, ethyl acetate (EtOAc), and n-butanol (n-BuOH). Extracts with higher enzymatic activity (EtOAc and n-BuOH) were analyzed for compound identification. The BuOH extract was subjected to Sephadex LH-20 with MeOH as the eluent to give four groups. The L1 group was analyzed by NMR, resulting in the identification of casuarictin (Casu) by comparison with literature data. For further use of extracts, they were solubilized in saline and 5% DMSO (dimethyl sulfoxide).

### 4.3. Purification of sPLA2 from Commercial Crotoxin

Samples (1 mg) of commercial crotoxin from Sigma Aldrich (C3737 Sigma, St. Louis, MO, USA) were dissolved in 250 μL of buffer A and duly clarified and applied to the chromatographic column (C5 column, semi analytical), which was pre-equilibrated with buffer A for 20 min. Elution of sPLA2 was performed with a continuous linear gradient of buffer B (66% acetonitrile in 0.1% TFA), and monitoring of the chromatographic profile was done at 280 nm. The samples were submitted to enzymatic kinetics to verify the activity of the fractions.

### 4.4. Biochemical Analysis

#### 4.4.1. Treatment of sPLA2 with Extracts and Mangrove Tannin

Incubation of sPLA2 with the tannin, BuOH, or EtOAc partitions was performed as described [11]. Initially the compounds were completely dissolved in DMSO with the final concentration in the solution never exceeding 5% by volume during incubation. An aliquot of the solution or tannin (400 μL of 0.1 mM solution) was added to 400 μL of purified sPLA2 of *C. durissus terrificus* (1 mg/mL). The mixture was then incubated for 90 min at room temperature, and aliquots of 200 μL were loaded onto a preparative reverse phase column to separate the treated enzyme (sPLA2:Casu or sPLA2:partition). After equilibrating the column with an HPLC A buffer (aqueous trifluoroacetic acid (TFA)), the samples were eluted using a discontinuous gradient of HPLC B buffer (acetonitrile in TFA) at a constant flow rate of 1.0 mL/min. The chromatographic run was monitored at 214 nm.

#### 4.4.2. Mass Spectrometry

The molecular masses of sPLA2 and sPLA2:Casu were determined by matrix-assisted laser desorption ionization-time-of-flight mass spectrometry (MALDI-TOF) using a Voyager-DE PRO MALDI-TOF mass spectrometer (Applied Biosystems, Thermo, Waltham, MA, USA). One microliter of sample (sPLA2 and sPLA2:Casu) with a concentration adjusted to 3 mg/mL in TFA was mixed with 2 μL of the matrix α-cyano-4-hydroxycinnamic acid (v/v). The matrix was prepared with acetonitrile and TFA (v/v). Ion masses were determined with an acceleration voltage of 25 kV, and the laser was operated at 2890 kJ/cm^2^ with a 300 ns delay and the linear analysis mode.

#### 4.4.3. Phospholipase A2 Assay

PLA2 activity was measured using the assay described by [4,11], adapted for 96-well plates. The standard assay mixture contained 200 μL of buffer (10 mM Tris-HCl, 10 mM CaCl_2_, 100 mM NaCl, pH 8.0), 20 µL of substrate, 20 µL of water, and 20 µL of PLA2 in a final volume of 260 µL. After the addition of PLA2 (20 μg), the mixture was incubated for up to 40 min at 37 °C, with the absorbance being read at 10 min intervals. The enzyme activity, expressed as the initial velocity of the reaction (Vo), was calculated based on the increase in absorbance after the desired time in min. All assays were conducted in triplicate, and the absorbance at 425 nm was measured using a SpectraMax 340 multiwell plate reader (Molecular Devices, San Jose, CA, USA). The results are expressed as the variation of Vo in comparison with the native enzymatic activity of native sPLA2. Each treatment was conducted for *n* = 12.

### 4.5. Pharmacological Assay and Antimicrobial Assays

#### 4.5.1. Paw Edema

A paw edema assay was performed using a previously described protocol [4,11]. Female swiss mice (23 g) were anesthetized and subsequently euthanized by cervical dislocation posterior to a paw edema induced by a single subplantar injection of sPLA2. Paw volumes were measured immediately before the injection of the samples and at selected time intervals thereafter (0, 30, 60, 180, and 240 min) using a hydroplethysmometer (model 7150, Ugo Basile, Monvalle, Italy). The results are expressed as the increase in paw volume (mL) calculated by subtracting the initial volume. Each treatment was conducted for *n* = 5.

#### 4.5.2. Evaluation of Myonecrosis

Myotoxic activity was evaluated by the measurement of plasma creatine kinase (CK) released from damaged muscle cells. For this, we used a commercial creatine kinase kit measurement CK-NAC kit (Laborlab, London, UK), as described elsewhere. In the right gastrocnemius muscle, 50 μL of the 0.5 mg/mL sPLA2 sample was injected while the control mice received only an equal volume of 0.15 M NaCl. After 3 h, the animals were anesthetized, and samples were collected from the abdominal cavity into tubes containing heparin as an anticoagulant. Plasma was stored at −10 °C for a maximum of 12 h before the assay. The level of CK was then determined with 40 μL of plasma, which was incubated for 3 min at 37 °C with 1.0 mL of the reagent according to the kit protocol. The resulting activity was expressed in U/L. Each treatment was conducted for *n* = 5.

#### 4.5.3. Electron Microscopy

The gastrocnemius muscle was fixed for 5 min with a solution consisting of 2.5% glutaraldehyde, 1% paraformaldehyde, and 0.01% sodium azide in 0.05 M sodium cacodylate at pH 7.2 and then removed. The specimens were further fixed in the same solution overnight, followed by post-fixation in 1% OsO_4_ in 0.1 M sodium cacodylate, dehydration, and infiltration with epoxy resin (Agar 100), in accordance with routine procedures. Ultrathin sections were cut with diamond knives in a Reichert Ultracut E microtome and contrasted with lead citrate and uranyl acetate before examination in a Zeiss CEM 902 electron microscope. 

#### 4.5.4. Antibacterial Activity

*X. axonopodis* pv. *passiflorae* (gram-negative) and *C. michiganensis* pv. *michiganensis* (gram-positive) bacterial strains were harvested from fresh agar plates and suspended in distilled sterilized water (A600 nm = 3 × 10^8^ CFU/mL). Aliquots of the bacterial suspension were diluted to 10^3^ CFU/mL and incubated with PLA2 and modified PLA2 (250 μg/mL) for 1 h at 28 °C. After incubation, survival was assayed on nutrient (Difco) plates (*n* = 5). Electron microscopic assessments for observation of morphological alterations of *X. axonopodis* pv. *passiflorae* were made for both the control samples and samples containing PLA2-treated bacteria (A600 nm = 3 × 10^8^ CFU/mL). Samples were fixed with 4% glutaraldehyde in a buffer A (0.1 M cacodylate buffer, pH 7.4) for 1 h at 4 °C, embedded in 3% low-gelling-temperature agarose (SeaPlaque, Thermo, Waltham, MA, USA) and stored for 12 h in a fixative at 4 °C. Agarose pellets were immersed in the fixative for 2 h at 25 °C, washed three times for 10 min with buffer A, and fixed with 1% osmium tetroxide (Agar Scientific Ltd, Essex, UK) in buffer A for 2 h at 25 °C. Sections were washed three times, dehydrated in increasing concentrations of ethanol and propylene oxide, and embedded in Epon resin (Agar Scientific, Essex, UK). Polymerization was performed at 60 °C for 48 h, and ultrathin sections were prepared with a Sorvall MT2 ultramicrotome., Du Pont Company, Clinical and Instrument System Division, Sorvall^®^ Products, Wilmington, DE 19898, USA).

The sections were placed on 5% collodion-coated 100-mesh grids, and stained with 4% uranyl acetate (Agar Scientific) for 15 min, followed by staining with 2.6% lead citrate (Agar Scientific) for 15 min. Samples were observed under a Hitachi 1100 transmission electron microscope operating at 100 kV (Hitachi Scientific Instruments, Berkshire, UK).

### 4.6. Statistical Analysis

The results are reported as the means ± SEM of the replicated experiments. The significance of the differences between the means was obtained by one- or two-way analysis of variance (ANOVA) followed by the Dunnett or Bonferroni test, when several experimental groups were compared with the control group. The confidence limit for significance was 5%.

## 5. Conclusions

In this paper, we isolated casuarictin (Casu), a tannin characterized for the first time in *L. racemosa*. This tannin showed anti-inflammatory capacity by neutralizing the enzymatic and pharmacological activity induced by sPLA2 from *C. durissus terrificus*. In addition, it was able to strongly inhibit edematogenic and myonecrotic activity. Likewise, the compound induced structural changes in sPLA2 and could also act as a source of external antioxidants. In relation to the antibacterial activity, this compound was able to interact with the bacterial membrane and induce the death of bacterial cells.

## Figures and Tables

**Figure 1 marinedrugs-17-00403-f001:**
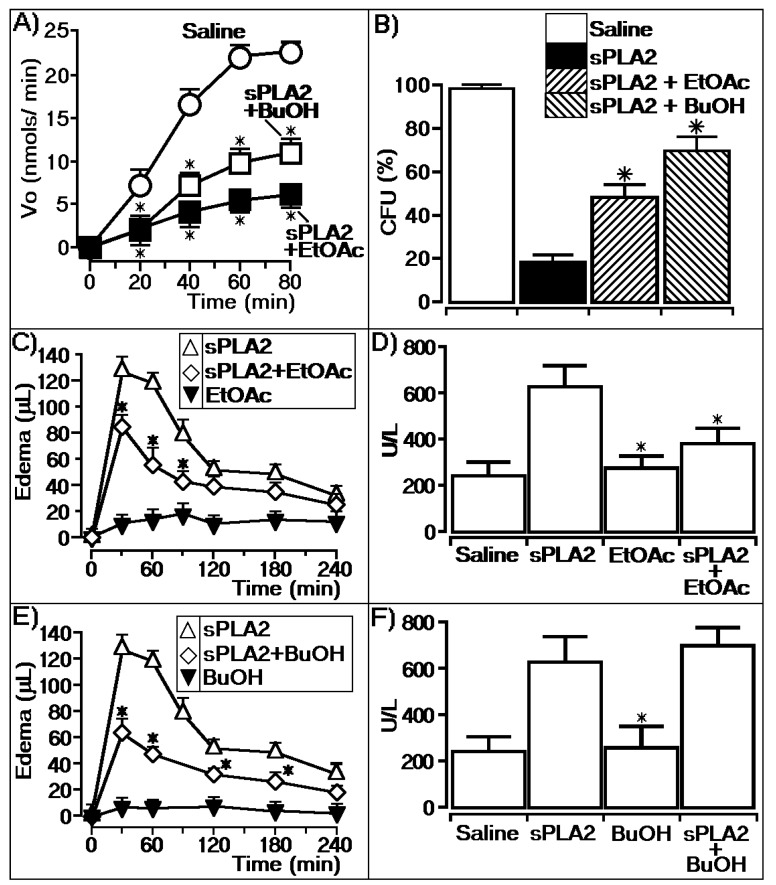
*Laguncularia racemosa* extracts and the enzymatic and pharmacological activities of secretory phospholipase A2 (sPLA2). (**A**) Activity of the buthanolic (BuOH) and ethyl acetate (EtOAc) phases on phospholipase A2 enzymatic activity, where saline represents the control group, BuOH represents sPLA2 treated with the buthanolic partition, and EtOAc represents sPLA2 treated with the ethyl acetate partition. The results are expressed as a variation of enzymatic velocity Vo, and the error bars indicate the SEM. * Statistically significant differences (*n* = 5, *p* < 0.05) compared to native sPLA2. (**B**) Effect of native sPLA2 and the proteins treated with the BuOH and EtOAc partitions against *Clavibacter michiganensis* pv. *michiganensis* (gram-positive bacteria). The error bars indicate the SEM of *n* = 6. * *p* < 0.05 compared with the saline control. (**C**,**E**) show paw edema induced after the injection of sPLA2 and sPLA2 treated with EtOAc and BuOH, respectively, into the right paws of Swiss mice. Edema is expressed as volume in µL and measurements were performed after 30, 60, 120, 180, and 240 min. The error bars indicate the SEM of five experiments. * *p* < 0.05 compared to native sPLA2. (**D**,**F**) show the myonecrosis assay and the results are expressed as creatine kinase (CK) units of enzymatic activity per liter (U/L). The error bars indicate the SEM of five experiments. * *p* < 0.05 compared with native sPLA2.

**Figure 2 marinedrugs-17-00403-f002:**
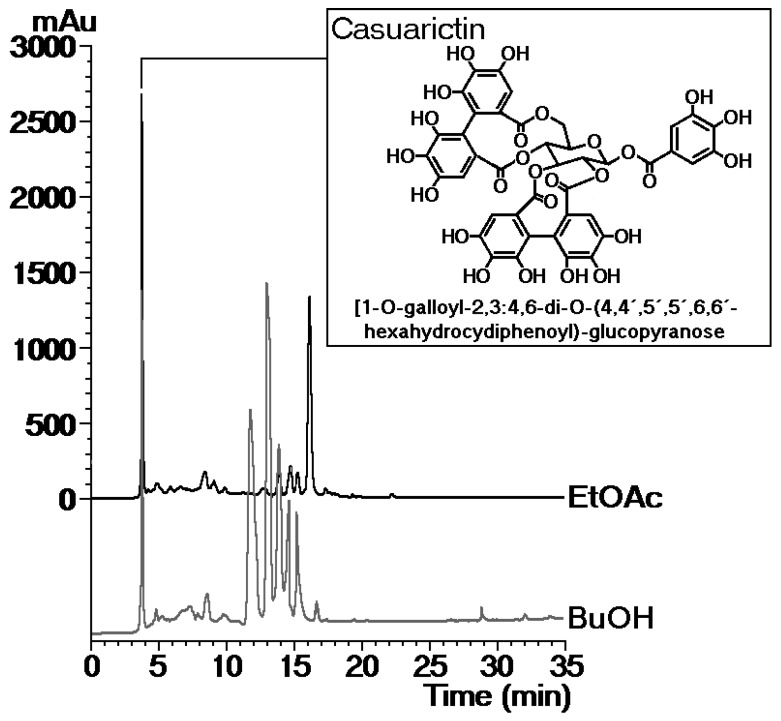
Chemical structure of casuarictin and chromatographic profiles of the EtOAc and BuOH phases obtained from the ethanolic extraction of *L. racemosa* leaves.

**Figure 3 marinedrugs-17-00403-f003:**
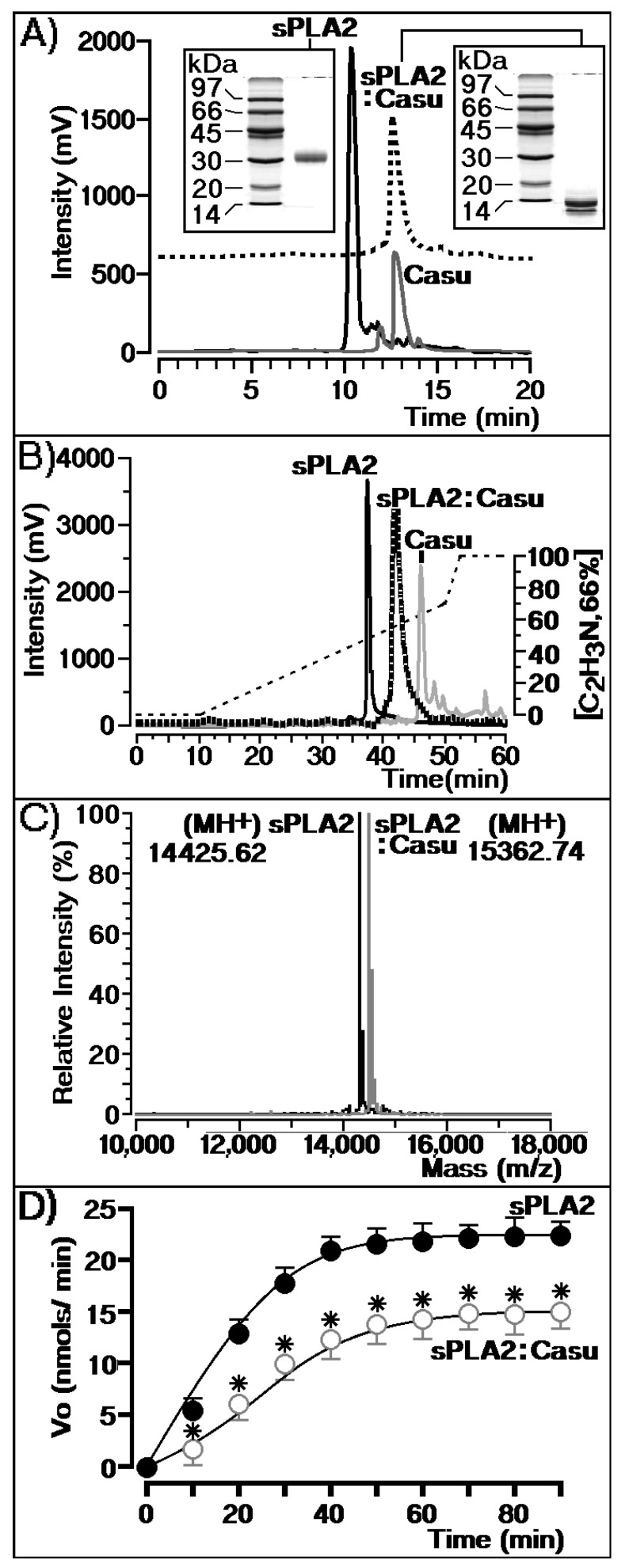
Chromatographic and enzymatic casuarictin (Casu) assays. (**A**) Chromatographic profiles of sPLA2, sPLA2 treated with Casu, and Casu alone. All fraction samples were analyzed under the same chromatographic conditions by size exclusion chromatography (SEC). (**B**) Shows the same samples treated with Casu subjected to reverse phase HPLC analysis. Both analyses provide some information on the molecular stability of Casu treated with sPLA2 that includes formation of the stable molecular complex sPLA2:Casu. (**C**) Shows the sPLA2:Casu complex and native sPLA2 with molecular masses of 15,362.74 and 14,425.62 Da, respectively. (**D**) Shows the enzymatic velocity of sPLA2 before and after treatment with Casu.

**Figure 4 marinedrugs-17-00403-f004:**
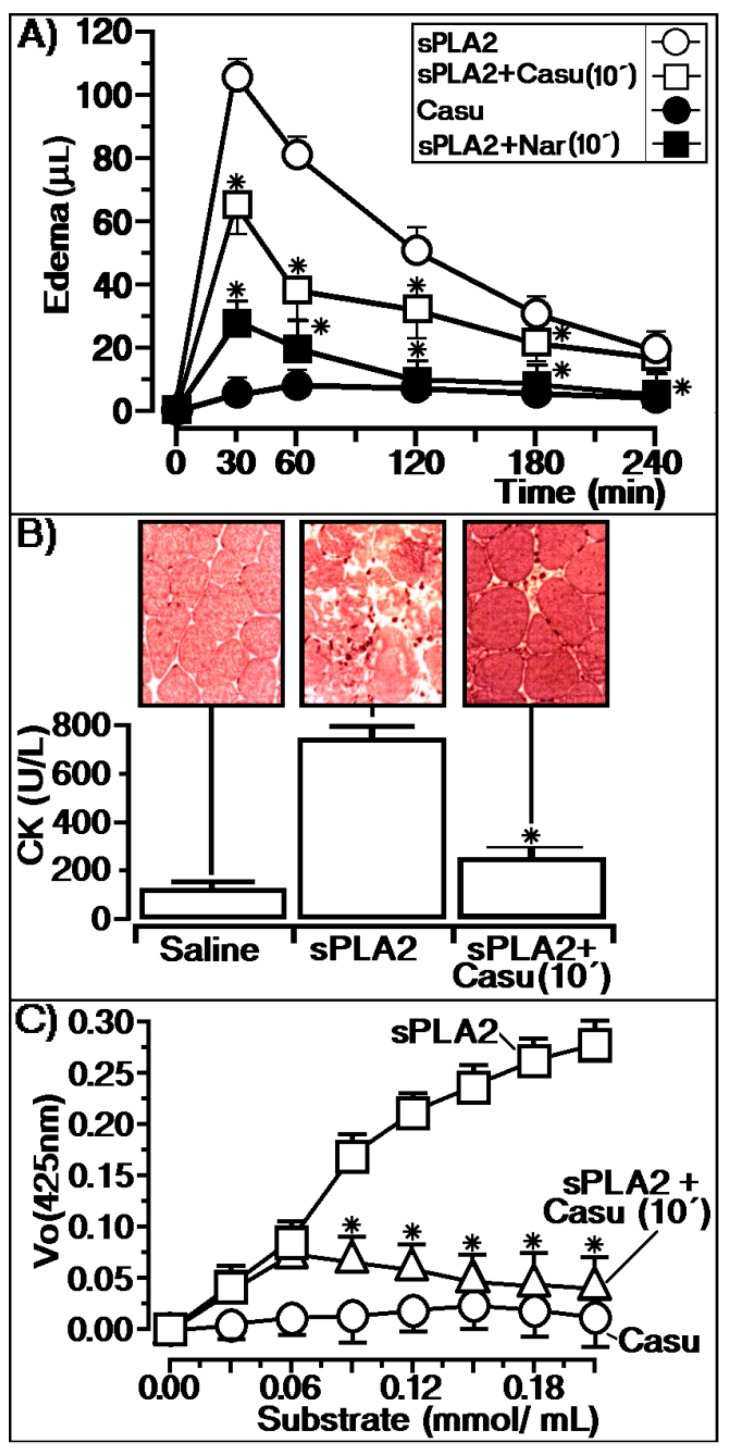
(**A**) Shows the protective effect of Casu injected into animals 10 min before the injection of native sPLA2 and edema induced by Casu alone. In (A), we also show the protective effect of naringenin (Nar) administered under the same conditions as Casu. This compound was used as an anti-inflammatory effect control. (**B**) Reveals the protective effective of Casu injected in the same manner as shown in (A). This figure shows that Casu injected 10 min after sPLA2 application neutralizes the myotoxic effects induced by sPLA2 once the CK levels decrease, revealing a histological similarity with saline. (**C**) Exhibits an irreversible profile of Casu, in which the enzymatic activity of sPLA2 is diminished in a dose-dependent manner.

**Figure 5 marinedrugs-17-00403-f005:**
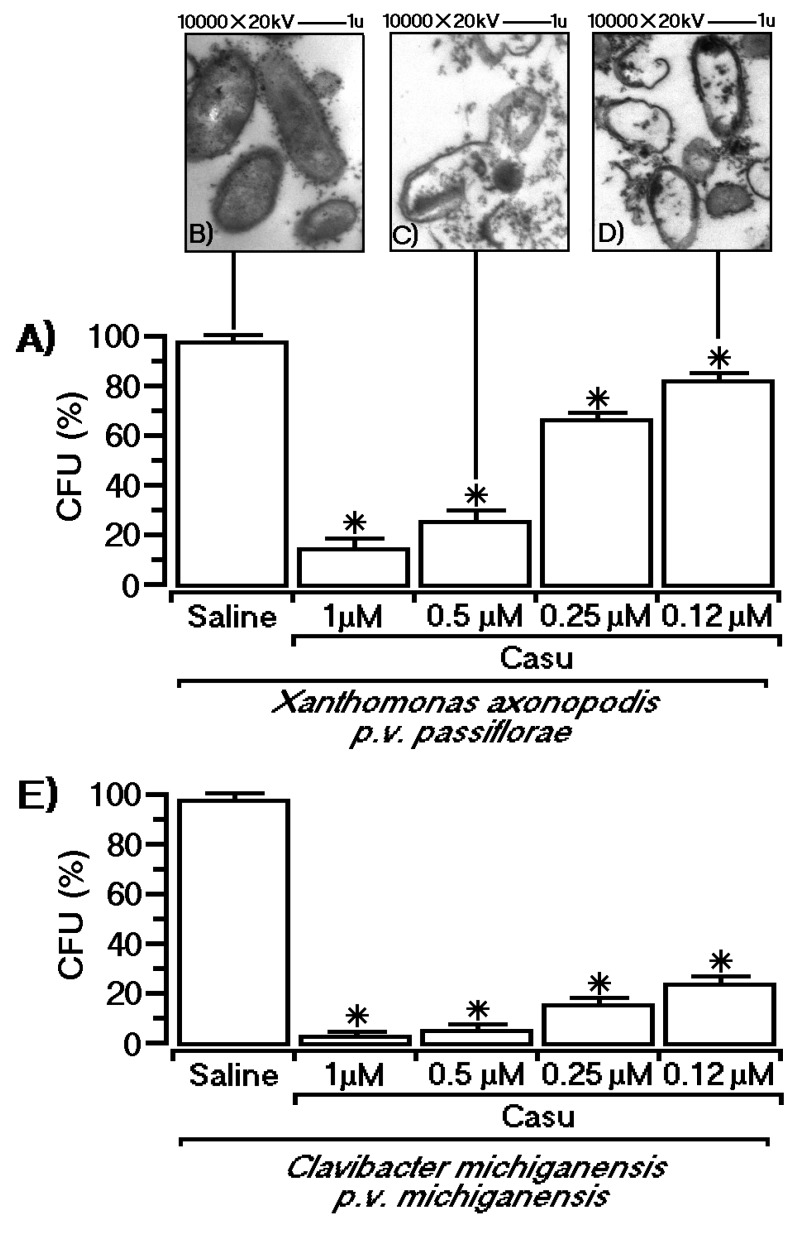
Antimicrobial effects of Casu against gram-positive and gram-negative phytobacteria. In both cases, antimicrobial effects occurred in a dose-dependent manner. The effect of Casu is shown against *Xanthomonas axonopodis* pv. *passiflorae* (**A**) and *C. michiganensis* pv. *Michiganensis* (**E**). Electron microscopy images show that Casu induces a massive membrane rupture, leading to a decrease in bacterial survival, in a dose dependent manner, as shown in (**B**–**D**). One-way ANOVA with Dunnett’s test was used, and values of *p* < 0.05 were considered significant as indicated by asterisk.

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
