# Peer review of "Evaluation of the Inhibitory Potential of Casuarictin, an Ellagitannin Isolated from White Mangrove (Laguncularia racemosa) Leaves, on Snake Venom Secretory Phospholipase A2"

_marinedrugs, 2019, doi:10.3390/md17070403_

Round 1

Reviewer 1 Report

This manuscript describes the isolation, structural elucidation, anti-inflammatory effect induced by sPLA2, their mechanisms, and antimicrobial activity of casuarictin from leaves of the species Laguncularia racemose. The article is well written, and the structures have convincingly established by providing proper experimental evidences. Overall, the novelty of the biological activity displayed by the compound is enough for publication in Marine Drugs. Nonetheless, some aspects need to be revised/changed before publication.

Point 1: Please mention the representative control compound with inhibiting inflammatory effects induced by sPLA2 in the manuscript for comparison with casuarictin. Based on the comparison of inhibitory activities of causarictin and control compound, the potency of casuarictin will be easily evaluated.

Point 2: (Line 204) Please describe which residue of causarictin acts on the catalytic site of sPLA2. It will be better to mention the author’s experimental evidences or related references.

Minor points

Point 3: This manuscript has lots of typing errors such as superscript, subscript, italic and spacing. For example Lane 43: Km2 -> km2, Lane 80: 1H NMR -> 1H NMR, Lane 196: in vitro and in vivo. -> in vitro and in vivo., Lane 257: H2O (1:2, v/ v) -> :H2O (1:2, v/v), Lane 291: CaCl2 -> CaCl2, etc. Thus, this manuscript should be edited to avoid English mistakes.

Point 4: Figure 2 showed that casuarictin and other peaks were detected in chromatographic profiles. Except of casuarictin, did the author evaluate enzymatic sPLA2 activities of other peaks? If the author did it, please mention results.

Point 5: (Line 89) Based on NMR data (1H, 13C), the author elucidated casuarictin structure and compared with literature data. Please mention used reference which compare with NMR data of casuarictin.

Point 6: References need to be consistent with the marine drugs journal reference style. Simple error is too many.

Point 7: remove ‘A)’ placed left upper of Figure 5.

Author Response

Reviewer 1

At first we appreciate the reviewer's prestigious collaboration and have done our best to meet your requests. As the article was reviewed by three reviewers, we focused on the first moment to meet the requests of each reviewer and after we have the final version of the article, we will contract the article correction and formatting services offered by the publisher. Thus,

Dear Reviewer 1, thanks for the kindly suggestions. We alter the sentences like you suggested.

This manuscript describes the isolation, structural elucidation, anti-inflammatory effect induced by sPLA2, their mechanisms, and antimicrobial activity of casuarictin from leaves of the species Laguncularia racemose. The article is well written, and the structures have convincingly established by providing proper experimental evidences. Overall, the novelty of the biological activity displayed by the compound is enough for publication in Marine Drugs. Nonetheless, some aspects need to be revised/changed before publication.

Point 1: Please mention the representative control compound with inhibiting inflammatory effects induced by sPLA2 in the manuscript for comparison with casuarictin. Based on the comparison of inhibitory activities of causarictin and control compound, the potency of casuarictin will be easily evaluated.

The control we use in many of our work is Naringin and we use it for internal control of our activities, even if they are not published. This is due to the fact that we have an extensive documentation of the anti-inflammatory activity of this compound and the fact that we know how this compound can interact with sPLA2.

Santos, ML., Toyama, D.O., Oliveira, S.C., Cotrim, C.A., Diz-Filho, E.B., Fagundes, F.H., Soares, V.C., Aparicio, R. and Toyama, M.H. 2011. Modulation of the pharmacological activities of secretory phospholipase A2 from Crotalus durissus cascavella induced by naringin. Molecules. 2011 Jan 18;16(1):738-61. doi: 10.3390/molecules16010738.

Mentions of the comparison were placed in the results and discussions.

Point 2: (Line 204) Please describe which residue of causarictin acts on the catalytic site of sPLA2. It will be better to mention the author’s experimental evidences or related references.

We hypothesize the loss of the ability of the substrate to access the catalytic site by the major reduction of the enzymatic activity when the PLA2 was incubated with Casu by a dose-dependent manner. We add the “in the enzymatic assay” in the phrase to emphasize that. Thank you for the suggestion.

Minor points

Point 3: This manuscript has lots of typing errors such as superscript, subscript, italic and spacing. For example Lane 43: Km2 -> km2, Lane 80: 1H NMR -> 1H NMR, Lane 196: in vitro and
in vivo. -> in vitro and in vivo., Lane 257: H2O (1:2, v/ v) -> :H2O (1:2, v/v), Lane 291: CaCl2 -> CaCl2, etc. Thus, this manuscript should be edited to avoid English mistakes.

We fixed the typing errors.

Point 4: Figure 2 showed that casuarictin and other peaks were detected in chromatographic profiles. Except of casuarictin, did the author evaluate enzymatic sPLA2 activities of other peaks? If the author did it, please mention results.

We don’t evaluate the activities of the other peaks.

Point 5: (Line 89) Based on NMR data (1H, 13C), the author elucidated casuarictin structure and compared with literature data. Please mention used reference which compare with NMR data of casuarictin.

We added the reference used to identify the casuarictin.

Point 6: References need to be consistent with the marine drugs journal reference style. Simple error is too many.

We changed the reference style. Sorry for the errors.

Point 7: remove ‘A)’ placed left upper of Figure 5.

The A) was removed from the figure 5.

Reviewer 2 Report

Manuscript "Evaluation of casuarictin an Ellagitannin isolated from Laguncularia racemosa leaves on snake venom secretory phospholipase A2" by Bittencourt Rodrigue et al. describes scientifically sound study of tannin named casuarictin activity towards snake venom phospholipase A2 and plant pathogenic bacteria Clavibacter michiganensis and Xanthomonas axonopodis. Methods are applied with scientific rigor. However, the manuscript (at least some parts of it) needs an extensive English editing.

In my view, the manuscript is interesting due to the wide spectrum of methods applied and importance of the snake envenomation issues caused by sPLA2.

15 "casuarictin (Casu) was isolated of some plant species" --> "casuarictin (Casu) was identified in some plant species"

15-16 "However, there are not any" would probably sound better if re-written smth like "To our knowledge no investigation of PLA2 inhibition by Casu has been performed yet"

16 "concerning to inhibition"

17-18 "from chromatographic procedures of" should be rephrased as "by chromatography"

21 "consisting OF", "Analysis"

26 "showed a potential therapeutic" --> to smth like "might be considered as a potential anti-venom compound inhibiting sPLA2"

In abstract it should be noted that Casu was injected 10 min after the sPLA2 injection because it supports the claim of potential utility of Casu as a snake venom antidote. In many articles claiming similar potential utility against snake toxins, "potential drugs" were used to pre-treat toxins in vitro prior to injection which is not particularly fair. In your article Casu performs well after the model "snake bite" event and it is (in my opinion) the strong point of the study.

49 "ecological role" --> "ecological importance/significance" probably sounds better

50 "flavonoids reveals to be able to inhibit" --> "flavonoids inhibit"

67-68 delete "In Figure 1A we observed that" and add "(Fig. 1A)" at the end of the sentence

Section 2.2: Please provide some explanation to the m\z difference between PLA2 and PLA2:Casu explicitly in the text. It is not clear what this difference (937 Da) means. Does it mean covalent modification of the PLA2?

In Fig. 5 legend please indicate what asterisc means and what statistical analysis was performed (criteria used to show significance). Also, delete "A)" from the figure itself.

Author Response

Reviewer 2

At first we appreciate the reviewer's prestigious collaboration and have done our best to meet your requests. As the article was reviewed by three reviewers, we focused on the first moment to meet the requests of each reviewer and after we have the final version of the article, we will contract the article correction and formatting services offered by the publisher. Thus,

Dear Reviewer 2, thank you for the suggestions, we improve the article based on that.

Manuscript "Evaluation of casuarictin an Ellagitannin isolated from Laguncularia racemosa leaves on snake venom secretory phospholipase A2" by Bittencourt Rodrigue et al. describes scientifically sound study of tannin named casuarictin activity towards snake venom phospholipase A2 and plant pathogenic bacteria Clavibacter michiganensis and Xanthomonas axonopodis. Methods are applied with scientific rigor. However, the manuscript (at least some parts of it) needs an extensive English editing.

In my view, the manuscript is interesting due to the wide spectrum of methods applied and importance of the snake envenomation issues caused by sPLA2.

15 "casuarictin (Casu) was isolated of some plant species" --> "casuarictin (Casu) was identified in some plant species"

We alter the sentence as you suggested

15-16 "However, there are not any" would probably sound better if re-written smth like "To our knowledge no investigation of PLA2 inhibition by Casu has been performed yet"

We modify the phrase, including your suggestion

16 "concerning to inhibition"

By adding the previous suggestion, we removed the phrase

17-18 "from chromatographic procedures of" should be rephrased as "by chromatography"

We changed the sentence

21 "consisting OF", "Analysis"

We add the of in the phrase

26 "showed a potential therapeutic" --> to smth like "might be considered as a potential anti-venom compound inhibiting sPLA2"

We changed the sentence based on your suggestions

In abstract it should be noted that Casu was injected 10 min after the sPLA2 injection because it supports the claim of potential utility of Casu as a snake venom antidote. In many articles claiming similar potential utility against snake toxins, "potential drugs" were used to pre-treat toxins in vitro prior to injection which is not particularly fair. In your article Casu performs well after the model "snake bite" event and it is (in my opinion) the strong point of the study.

Thank you for the suggestion, we put the method in the abstract to emphasize the potential of the study.

49 "ecological role" --> "ecological importance/significance" probably sounds better

We modify the phrase

50 "flavonoids reveals to be able to inhibit" --> "flavonoids inhibit"

We changed the sentence.

67-68 delete "In Figure 1A we observed that" and add "(Fig. 1A)" at the end of the sentence

The alteration was done.

Section 2.2: Please provide some explanation to the m\z difference between PLA2 and PLA2:Casu explicitly in the text. It is not clear what this difference (937 Da) means. Does it mean covalent modification of the PLA2?

The results of the analyzes between sPLA2 and sPLA2: Casu show a difference of 937 Da, which is the molecular mass of the tannin. These results show that the tannin Casu was able to form a stable complex with sPLA2 and that probably involves strong non-covalent interactions such as hydrogen bonds and hydrophobic interactions.

The modification was stable after the reverse phase treatment, so, probably means a stable modification, but we didn’t affirm the covalent modification by the methods applied here.

In Fig. 5 legend please indicate what asterisc means and what statistical analysis was performed (criteria used to show significance). Also, delete "A)" from the figure itself.

We modify the figure and the caption to inform the stastitical analysis performed there.

Reviewer 3 Report

The whole manuscript need to revise for correcting grammatical error. The purpose of this study need to be clearly written.

Title: Evaluation of what aspect? Title is not clear, please rewrite

Line 3: Write down the common name of Laguncularia racemosa

Line 15: Please check the grammar.... You can write like this way, no study was conducted regarding

Line 19: Please write down the elaborated form of sPLA2 first and then within bracket you can put it abbreviated form. Write down the common name of Crotalus durissus terrificus.

Line 25: Showed significant antimicrobial activity (delete a)

Line 25:  In addition, this compound also showed a significant antimicrobial activity. Therefore, 25 Casu showed a potential therapeutic application on edema and myonecrosis induced by snake venom 26 secretory phospholipase A2 and antimicrobial effect. (These lines were already mentioned above, therefore, not necessary).

Line 32: Phospholipase A2 not phospholipases A2

Line 32 to 34: Please rewrite the grammar and sentence structure not ok.

Line 44: Instead of found use available

Line 48: revealed not reveal

Line 53: Check spelling of diarrhea

Line 68: Please write down it was observed that

Line 73: showed that not show that

Please clarify at the beginning, what you mean by pharmacological activity.

Line 77: showed that

Line 184: revealed

Line 192: Please check the sentence (however, ....is essential)

Author Response

Reviewer 3

At first we appreciate the reviewer's prestigious collaboration and have done our best to meet your requests. As the article was reviewed by three reviewers, we focused on the first moment to meet the requests of each reviewer and after we have the final version of the article, we will contract the article correction and formatting services offered by the publisher. Thus,

Thanks for the suggestions, we modify the text according to your suggestions.

The whole manuscript need to revise for correcting grammatical error. The purpose of this study need to be clearly written.

Title: Evaluation of what aspect? Title is not clear, please rewrite

We modify the title including “ the inhibitory potential of” after the “Evaluation”.

Line 3: Write down the common name of Laguncularia racemosa

We add the common name of the plant

Line 15: Please check the grammar.... You can write like this way, no study was conducted regarding

We alter the phrase based on your and the Reviewer 2 suggestions.

Line 19: Please write down the elaborated form of sPLA2 first and then within bracket you can put it abbreviated form. Write down the common name of Crotalus durissus terrificus.

We add the information’s on the sentence.

Line 25: Showed significant antimicrobial activity (delete a)

We delete the “a”

Line 25: In addition, this compound also showed a significant antimicrobial activity. Therefore, 25 Casu showed a potential therapeutic application on edema and myonecrosis induced by snake venom 26 secretory phospholipase A2 and antimicrobial effect. (These lines were already mentioned above, therefore, not necessary).

We delete the first sentence

Line 32: Phospholipase A2 not phospholipases A2

We delete the plural

Line 32 to 34: Please rewrite the grammar and sentence structure not ok.

We rewrited

Line 44: Instead of found use available

We changed the word according to your suggestion

Line 48: revealed not reveal

We alter the word.

Line 53: Check spelling of diarrhea

We checked the spelling and it was right.

Line 68: Please write down it was observed that

We changed the sentence

Line 73: showed that not show that

We alter the word and delete the “that”.

Please clarify at the beginning, what you mean by pharmacological activity.

We consider the pharmacological activity the in vitro activities. So, after describing the inhibition, we add a phrase that says “which means, the in vivo activities”

Line 77: showed that

We fix the sentence

Line 184: revealed

We changed the word

Line 192: Please check the sentence (however, ....is essential)

We checked the sentence and modify to a better understanding.

Round 2

Reviewer 1 Report

The authors were cleared and revised to reviewer's comments and suggestions.

Reviewer 3 Report

Much improved.